# The Impact of Exercise on Immunity, Metabolism, and Atherosclerosis

**DOI:** 10.3390/ijms24043394

**Published:** 2023-02-08

**Authors:** Ulrike Meyer-Lindemann, Aldo Moggio, Alexander Dutsch, Thorsten Kessler, Hendrik B. Sager

**Affiliations:** 1Department of Cardiology, German Heart Center Munich, Technical University Munich, 80636 Munich, Germany; 2DZHK e.V. (German Centre for Cardiovascular Research), Partner Site Munich Heart Alliance, 80336 Munich, Germany

**Keywords:** exercise, inflammation, metabolism, physical activity, exercise immunology, atherosclerosis

## Abstract

Physical exercise represents an effective preventive and therapeutic strategy beneficially modifying the course of multiple diseases. The protective mechanisms of exercise are manifold; primarily, they are elicited by alterations in metabolic and inflammatory pathways. Exercise intensity and duration strongly influence the provoked response. This narrative review aims to provide comprehensive up-to-date insights into the beneficial effects of physical exercise by illustrating the impact of moderate and vigorous exercise on innate and adaptive immunity. Specifically, we describe qualitative and quantitative changes in different leukocyte subsets while distinguishing between acute and chronic exercise effects. Further, we elaborate on how exercise modifies the progression of atherosclerosis, the leading cause of death worldwide, representing a prime example of a disease triggered by metabolic and inflammatory pathways. Here, we describe how exercise counteracts causal contributors and thereby improves outcomes. In addition, we identify gaps that still need to be addressed in the future.

## 1. Introduction

A sedentary lifestyle is closely linked with increased incidence of cardiovascular and metabolic risk factors. In contrast, regular exercise decreases all-cause and cardiovascular morbidity and mortality as well as the incidence of type 2 diabetes [1,2]. Evidence also shows that exercise not only protects against communicable diseases including viral and bacterial infections, but also beneficially modifies the course of non-communicable diseases, such as cancer [3,4].

Most available data indicate that these beneficial effects rest on the metabolic and anti-inflammatory effects of exercise [5]. Indeed, it is known that physical exercise lowers cardiovascular risk by improving metabolic balance and preventing obesity [6]. The positive effects of exercise are also associated with an induction of an anti-inflammatory response. Particularly, cells of the immune system are highly responsive to exercise and undergo numerical and phenotypical changes [7]. However, depending on exercise intensity and duration, exercise-primed immune cells react differently.

“Exercise” refers to planned, structured, repetitive, and intentional movement intending to improve or maintain fitness [8]. According to the guidelines from the European Society of Cardiology (ESC), exercise can be classified based on absolute and relative intensity. Light and moderate exercise, defined as “somewhat-hard” on the Borg Rating of Perceived Exertion (RPE) scale, refers to activities including walking, playing golf, or performing water aerobics. During the activity, the heart rate should not exceed 77% of the age-related maximum (220-age). All intensities above are classified as vigorous exercise, referred to as “hard-really hard” on RPE scale. Activities such as running, swimming laps, or heavy gardening belong to the spectrum. Here, the heart rate may rise up to 95% of the maximum [9]. The term “physical activity” is less precisely defined and refers to any movement of muscles for which energy is required [8].

In this narrative review, we aim to highlight the beneficial effects of exercise on immunity and metabolism while discriminating between moderate and vigorous exercise.

We then consider how exercise beneficially modifies the course of atherosclerosis—a chronic inflammatory disease of the vessel wall. Identification of these mechanisms is critical for the implementation of guidelines and the development of drugs that mimic the effects of exercise. Our goal is to give up-to-date insight into the field of exercise and to provide an impetus for future research directions.

## 2. Exercise and Its Effects on Immunity

### 2.1. Moderate Exercise and Its Impact on the Immune System

#### 2.1.1. Acute Effects

Exercise induces an increased oxygen demand in muscles, which can be redeemed by increased cardiac output and greater blood flow, stimulated by the sympathetic nervous system [10]. Hemodynamic shear stress, as well as the release of catecholamines and glucocorticoids, force leukocyte release from the vascular, pulmonary, hepatic, and splenic reservoirs and subsequently increase circulating leukocyte numbers [11,12]. Thus, acute moderate exercise induces a leukocytosis in the blood, primarily driven by lymphocytosis and granulocytosis. The amount of leukocytosis positively correlates with the intensity and duration of exercise and is independent of individual fitness levels [7,13]. In the post-exercise period, the return to baseline levels varies between different leukocyte subsets. While the blood count of natural killer (NK) cells, a subset of lymphoid cells, decreases after only one hour [14], neutrophil counts return back to baseline rather delayed after 24 h [7,13].

Phenotypical changes in leukocytes induced by moderate exercise have not been investigated extensively. Still, it has been reported that moderate exercise leads to a transient upregulation of the intracellular vitamin D receptor (VDR) on T lymphocytes [15]. Vitamin D, essential for T cell development, differentiation, and effector functions, plays a key role in immunomodulation. Initiated effects include T cell phenotype switching by suppressing pro-inflammatory T_H_1 and T_H_17 cells and promoting regulatory T cells (T_reg_) [16]. By transiently inducing the expression of VDR, moderate exercise represents a potential adjuvant in the treatment of vitamin D deficiency, a disease with a worldwide prevalence of one billion [17] that is associated with several autoimmune diseases, including diabetes and multiple sclerosis, and increased susceptibility to viral infections [18].

Moreover, moderate exercise promotes neutrophil activation and migration. During their departure from the circulation to the site of action, neutrophils undergo phenotypical changes required for pre-activation. Here, the integrin α-M (also known as CD11b) is upregulated, while the surface receptor CD62 is cleaved off by proteolytic shedding [19,20]. Physical exercise enhances the expression of CD11b in neutrophils acutely, as an immediate response to exercise [7], but also chronically, as individuals who exercise regularly report increased levels of CD11b [21].

#### 2.1.2. Chronic Effects

In the long term, exercise particularly affects lymphocytes.

Immunosurveillance describes the ability of the immune system to recognize and fight against foreign pathogens, a process largely executed by patrolling CD8+ lymphocytes and NK cells [22,23]. After recognizing mutation-derived neoantigens, CD8+ cytotoxic cells induce apoptosis of malignant cells through the release of cytotoxic substances and activation of the death receptor [24]. The corresponding cells in innate immunity are NK cells [25]. To perform, CD8+ cells and NK cells must arrive at the site of action. Recently, it was reported that physical exercise improves the trafficking behavior of both CD8+ cells and NK cells. Moderate exercise increases the number of CD8+ cells expressing C-X-C motif chemokine receptor 3 (CXCR3) [26,27]. CXCR3 is an inflammatory chemokine receptor orchestrating the migration of activated T cells in the inflamed periphery [28]. Further, the increased release of catecholamines during exercise promotes the mobilization of NK cells while skeletal muscle-derived IL-6 encourages the redistribution of NK cells leading to an increased uptake in tumors [29,30]. The improved recruitment capacities of cytotoxic CD8+ cells and NK cells correlate with decreased tumor size [26,27] and improved survival in cancer patients [31]. Strikingly, the anti-neoplastic effects of exercise seem to be persistent: adoptively transferred CD8+ cells from exercising mice caused enhanced survival and reduced tumor growth in mice [26]. The recruitment profile of B lymphocytes remains unaffected by exercise [30].

Additionally, regular physical exercise counters immunosenescence, the progressive age-related deterioration of the immune system. The age-related atrophy of the thymus, the organ primarily responsible for T cell homeostasis, leads to a reduced supply of naïve T lymphocytes, while antigen-specific memory T cells relatively increase [32]. Thus, T cells have an attenuated capability to interact with novel pathogens, to adequately respond to vaccinations, and to repress latent infections, such as varicella-zoster [33]. Moderate exercise, however, may ward off this process by preserving the thymic output. Increased frequencies of naïve T lymphocytes, less pronounced expansion of memory cells, and less accumulation of senescent cells have been reported in physically active older adults, demonstrating the exercise-induced “rejuvenation” of the immune system [13,34,35]. Thereby, exercise beneficially modifies the course of several inflammatory diseases because chronic inflammation and senescent cells are mutually dependent. By constantly activating immune cells, chronic inflammation promotes cellular senescence. Vice versa, senescent cells produce pro-inflammatory cytokines and foster inflammation [36,37]. Complementary, physical exercise helps to maintain a regular CD4/CD8 ratio [38] and elevates the number of immunomodulatory T_REG_ cells [38,39]. On the other hand, exercise does not cause the preservation of naïve B lymphocytes [34]. Finally, no differences in surface receptors of neutrophils have been reported as a chronic effect of moderate exercise [21,40].

### 2.2. Vigorous Exercise and Its Impact on the Immune System

#### 2.2.1. Acute Effects

As exercise-induced leukocytosis positively correlates with the intensity and duration of an activity, the amount of released cells during acute vigorous exercise is more pronounced [7,13]. Alterations in lymphocyte numbers are well studied and follow a biphasic course. First, exercise induces an instantaneous lymphocytosis lasting 45–60 min [41]. The release of catecholamines during exercise provokes a tremendous increase in NK cells (10×) and CD8+ T lymphocytes (2.5×) via ß2-adrenergic receptor (ß2-AR) signaling [4,42,43,44]. Mobilization of CD4+ cells occurs independently from ß2-AR and is less pronounced. Contrary to the effector status regulated release of T lymphocytes, the nearly twofold increase in B lymphocytes is mainly due to the enhanced release of immature subsets [45]. The pronounced increase in lymphocyte numbers is followed by a massive decrease within one to three hours post-exercise provoking a transient lymphopenia [13,39,46,47], which is primarily driven by a reduction in NK cells and CD8+ T lymphocytes [48]. This induced lymphopenia used to be commonly seen as part of a short-term suppression of the immune system leading to an increased susceptibility for infections, a phenomenon called the “open-window” theory [49]. However, the contemporary opinion is that transient lymphopenia is only a snapshot of the increased redeployment of lymphocytes promoting immunosurveillance and immunoregulation [4,12]. Another contributing factor to exercise-induced lymphopenia is enhanced apoptosis of highly differentiated T cells in the early post-exercise period, characterized by increased expression of the death receptor CD95 [50,51]. The induced gap is replenished by increased mobilization of naïve T cells indicating a regenerative effect of exercise [52].

Exercise does not only encourage the mobilization of lymphoid, but also of myeloid cells. The exercise-induced monocytosis is, similar to the lymphocytosis, highly dependent on the sensitivity to ß2-AR signaling and monocyte subsets; classical (CD14++ CD16-), non-classical (CD14+ CD16++), and intermediate (CD14+ CD16+) monocytes [53,54] respond differently. An acute bout of exercise preferably induces an increase in non-classical monocytes [44,55] while the rise of classical and intermediate monocytes is less pronounced [44]. The count of circulating monocytes rapidly returns back to baseline within one hour after exercise [55,56]. In contrast to the fast decline of lymphocytes and monocytes, neutrophil numbers continue to rise for one hour after exercise cessation and return to baseline delayed after 24 h [47,51,57].

By phenotypic changes, vigorous exercise creates an anti-inflammatory milieu. Vigorous exercise acutely elevates the expression of CD39 and CD73 on CD4+ T lymphocytes [58]. These surface-associated enzymes catalyze the conversion of adenosine triphosphate (ATP)/adenosine diphosphate (ADP) to adenosine monophosphate (AMP)/adenosine, respectively. While ATP promotes inflammation, fulfilling its role as a “danger” signal in inflammation, the nucleotide promotes the migration of phagocytes to damaged tissues, adenosine triggers a series of anti-inflammatory responses [39,59]. By promoting the degradation of ATP to adenosine via the CD39/CD73 pathway, vigorous exercise induces a shift toward an anti-inflammatory environment [58]. Further, vigorous exercise reduces the expression of Toll-like receptors (TLRs) 1, 2, and 4 on CD14+ monocytes in the early postexercise period [60]. Cells of innate immunity are equipped with TLR to rapidly fight against invading pathogens. However, excess activation of TLR leads to the sustained secretion of pro-inflammatory cytokines and chemokines which contributes to the progression of numerous inflammatory and auto-immune diseases [61,62]. By reducing the expression of TLR, exercise provokes anti-inflammatory effects which may beneficially alter the course of diseases, such as atherosclerosis, asthma, or systemic lupus erythematosus [63].

#### 2.2.2. Chronic Effects

In the long term, too, exercise promotes anti-inflammatory effects.

Vigorous exercise encourages an anti-inflammatory phenotype switching in lymphocytes and monocytes. The balance between T_H_1 and T_H_2 cells shifts in favor of T_H_2 [46,47]. Within the monocyte population, vigorous exercise decreases the frequency of intermediate and non-classical monocytes by diminishing the expression of CD16 [64,65]. Similar to moderate exercise, vigorous exercise leads to a downregulation of TLR [65]. Thus, the age-associated decline in monocyte function, characterized by increased expression of CD16 and impaired TLR expression [66,67], might be counteracted by exercise. Similar to moderate exercise, vigorous exercise preserves thymic output. VO_2_, the maximal oxygen uptake during physical exertion, positively correlates with the extent of naïve T lymphocytes [68] and the number of Interleukin-10 (IL-10)-producing T_REG_ cells [39,47]. Finally, no difference in surface receptors of neutrophils has been reported [21,40].

In conclusion, exercise promotes mainly anti-inflammatory effects and hence may primarily exert advantageous effects in diseases with strong inflammatory components. Consequently, we now focus on how physical exercise potentially impacts the development and progression of atherosclerosis, a chronic inflammatory disease of the vessel wall. The multiple effects of exercise on immunity are summarized in Figure 1.

## 3. Exercise and Its Effects on Atherosclerosis

Atherosclerosis is a chronic inflammatory disease of large- and medium-sized arterial blood vessels with the formation of lipid-containing plaques in the intima. Consequences of these atheromatous plaques include progressive plaque growth leading to narrowing of the vessel lumen and plaque erosion/rupture causing atherothrombosis with total luminal occlusion, which may result in life-threatening cardiovascular events, such as myocardial infarction (MI) or stroke [69,70].

Pathogenetically, atherosclerosis is an interplay of metabolic and inflammatory pathways that are indispensably linked. High plasma lipid levels, as well as hemodynamic shear stress, cause functional impairment of the endothelium, the innermost layer of the vessel wall. This impaired function of endothelial cells (ECs) allows subendothelial accumulation of low-density lipoprotein (LDL)-derived cholesterol. Additionally, it promotes the recruitment of inflammatory cells into the vessel wall [71,72,73], which can be enhanced by different lifestyle-associated risk factors [74,75]. Here, monocytes, neutrophils, and T lymphocytes create a pro-inflammatory, disease-propagating environment that contributes to all stages of the disease [72].

The close link between metabolic and inflammatory causes becomes apparent through the formation of foam cells. Once inside the vessel wall, monocytes differentiate into macrophages, the major leukocyte population in plaques [76], strongly engulf LDL and develop into foam cells. The accumulation of foam cells in the vessel wall indicates the formation of fatty streaks, a hallmark of early-stage lesions [69]. Furthermore, the generation of foam cells promotes the proliferation and synthesis of extracellular matrix components in subendothelial cells promoting plaque expansion [77,78].

Metabolic-associated risk factors play a crucial role in the initiation of atherosclerosis. The metabolic syndrome refers to the co-occurrence of the most important risk factors including abdominal obesity, hypertension, impaired glucose tolerance/diabetes mellitus, and atherogenic dyslipidemia [79,80]. The underlying mechanism is an excess of energy caused by a hypercaloric diet (=increased energy intake) and lack of physical activity (=decreased energy expenditure). The surplus, non-usable energy is stored and excess carbohydrates are saved as glycogen in skeletal muscle and liver, while redundant fatty acids are esterified with glycerol to build triacylglycerol (TAG), deposited in muscle and adipose tissue. If needed, the organism can catabolize these metabolic fuels to generate ATP, the main energy provider in human cells. In metabolic syndrome, however, there is a permanent energy surplus. The energy reserves are hardly touched but they are filled up, further perturbating the physiological metabolism in numerous ways.

Exercise, however, increases energy consumption. The working, contracting muscle relies on ATP, the main energy provider in human cells, as it is required for the function of several key enzymes involved in membrane excitability (Na+ K+ ATPase), myofilament cross-bridge cycling (myosin ATPase), and muscle relaxation (sarco/endoplasmic reticulum Ca2+ ATPase) [81]. Intramuscular ATP stores are relatively small and rapidly depleted [82]. Thus, enduring exercise requires the promotion of ATP-generating pathways. The choice of metabolic fuel, i.e., carbohydrate or fat, depends on intensity, duration, and type of exercise [81].

Thus, exercise modifies CVD through the modulation of pivotal risk factors. We now address the multiple pathways by which exercise beneficially alters lipid and glucose metabolism, as well as inflammation. A summary of the most relevant human and animal trials used for this purpose is in Table 1 and Table 2, respectively.

### 3.1. The Effect of Exercise on Risk Factors for Atherosclerosis

#### 3.1.1. Lipid Metabolism

Fat primarily serves as a metabolic fuel for long-duration exercise of low intensity. Maximal rates of fat burning occur at 60–65% VO_2_ max [85]. The ATP-generating pathway, ß-oxidation, must be fueled with fatty acids. This requires the hydrolysis of TAG from adipose tissues, muscle, and plasma and the delivery of the released fatty acids to skeletal muscle mitochondria, where ß-oxidation takes place. Due to their spatial proximity, intramuscular TAG is tapped first, followed by TAG stored in adipose tissue, the largest “fuel reserve” in the body [107]. In the latter, hormone-sensitive lipase (HSL) catalyzes the rate-determining step, which is the cleavage of TAG into fatty acids and glycerol. Plasma-derived TAG packed in lipoproteins are hydrolyzed by the enzyme lipoprotein lipase (LPL) at the extracellular muscle membrane to enable intramuscular import [108].

Combining the results from 12 small-scaled studies revealed that a single session of exercise at an intensity of 50–80% VO_2_ max lowers circulating TAG levels by 3–15% for up to 72 h [109]. Primarily, exercise succeeds in TAG reduction by increasing metabolism. Mediated by ß-adrenergic stimulation [110,111], the rate of lipolysis is increased two- to threefold during exercise [83,84]. Indeed, lipolysis exceeds the uptake of fatty acids during exercise which transiently elevates the concentration of fatty acids in plasma [85,86]. Further, the rate of ß-oxidation, also stimulated by ß-receptor signaling, may be increased five- to tenfold [112]. Secondly, regular exercise (=training) induces several adaptive transformations for faster and more efficient energy production. In skeletal muscle, exercise induces capillary growth facilitating fatty acid delivery [87]. Further, the density of mitochondria is increased in physically active individuals, extending the capacity for ß-oxidation [113].

Hormone-sensitive lipase (HSL) is regulated in the short term by phosphorylation-dephosphorylation, primarily by the ß-adrenergic activation of cAMP-dependent protein kinase [114]. Exercise increases the sensitivity of HSL to catecholamines in intra-abdominal adipose tissue, while also increasing the amount of HSL [103,104]. Contrary to adipose tissue, the amount of HSL in skeletal muscle remains unaffected by exercise [103]. Thus, exercise predominantly facilitates the conversion of TAG to free fatty acids in adipose tissue.

Interestingly, the timing of exercise seems to determine the power of adipocyte lipolysis. The circadian clock, an internal timekeeping system, regulates multiple physiological processes by inducing 24 h rhythms in gene expression. The expression of HSL depends on two of these core clock genes, namely *Bmal1* and *Per2,* thereby resulting in a circadian variation of lipolysis [115,116,117]. Recently, it was observed that rats exercising during the late part of their active phase gained a stronger lipolytic potency compared to rats exercising in the early part because of increased sensitivity to stimulating ß-adrenergic signaling. Consequently, enhanced lipolysis during late-phase activity led to increased weight loss [104].

To date, it is unclear whether exercise affects the activity of LPL, the enzyme responsible for the intramuscular uptake of TAG. On the one hand, there are studies denying any increase during activity [89,118]. Contrarily, there are studies claiming the opposite [90]. It is noteworthy that the effect of elevated LPL activity disappears as soon as the exercise-induced energy deficit is replenished by food uptake. Therefore, increased LPL activity might be dependent on energy expenditure and simply portray a condition of calorie deficit [91].

While increased TAG levels are associated with higher cardiovascular risk, increased high-density lipoprotein (HDL) levels protect against cardiovascular disease (CVD) [119]. HDL is often referred to as the “good” cholesterol, fulfilling its function as a reverse cholesterol transporter [120]. It is well-established that both acute and chronic exercise increase HDL concentration in plasma in a dose-response manner, in which exercise volume is more determining than exercise intensity [109,121,122]. Still, the mechanism of HDL elevation after exercise is only partially understood.

One possible explanation is the exercise-induced modulation of peroxisome proliferator-activated receptors (PPARs). PPARs are a subfamily of ligand-activated nuclear receptors/transcription factors regulating the expression of various genes related to lipid and glucose metabolism [123]. Dysregulation of PPARs contributes to the pathogenesis of metabolic diseases, such as atherosclerosis [124]. Thus, in the treatment of these diseases, PPAR represents a pharmacological target that is, indeed, already addressed in clinical practice. Activation of PPARα by fibrates successfully reduces TAG and increases HDL levels [125,126], while activation of PPARγ by thiazolidinediones (TZDs)/glitazones may improve insulin sensitivity in patients with diabetes mellitus [127,128]. It has been shown that exercise engages in PPAR activation by increasing the expression of the *PGC-1α* gene [129]. PGC-1α, the master regulator of mitochondrial biogenesis, is a co-activator of PPARγ, permitting the interaction of the nuclear receptor with several transcription factors [130,131]. In conjunction with the finding that acute exercise directly provokes increased expression of PPARγ in skeletal muscles [92], exercise-induced activation of PPAR receptors causally contributes to improved lipid levels [132].

Further, exercise increases the activity of lecithin–cholesterol acyltransferase (LCAT) [93,133]. LCAT is located on the surface of precursor HDL (pre-ß-HDL) and catalyzes the esterification of cholesterol, which is an important step for reverse cholesterol transport [134]. Subsequently, the esterified cholesterol is translocated into the hydrophobic core of HDL, exposing new binding capacities for cholesterol at the surface. The increased incorporation of cholesterol esters in the core of the particle promotes the maturation of pre-ß-HDL to HDL.

Additionally, exercise promotes the biosynthesis of carnitine, an endogenous molecule enabling mitochondrial import and ß-oxidation of long-chain fatty acids [135,136]. Obesity and diabetes lead to carnitine insufficiency, which results in improper mitochondrial fuel metabolism with low rates of fatty oxidation and incomplete ß-oxidation [137]. Exercise, however, may rescue carnitine biosynthesis, as it was recently shown that exercise reversed the impaired carnitine status in mice caused by a high-cholesterol diet [105].

#### 3.1.2. Glucose Metabolism

Carbohydrates primarily serve as metabolic fuel for high-intensity exercise, with the greatest consumption occurring at 80–100% VO_2_ max [81]. The ATP-generating degradation of glucose (=glycolysis) takes place in the cytosol. The primary conversion of glucose into pyruvate generates two molecules of ATP. Further utilization of pyruvate is dependent on the oxygen supply of the cell. Under aerobic conditions, pyruvate is metabolized in the citric acid cycle (CAC). This process, called oxidative phosphorylation, produces 30 molecules of ATP. Required substrates are reducing equivalents (nicotinamide adenine dinucleotide hydrogen (NADH), flavin adenine dinucleotide hydrogen (FADH_2_), free ADP, phosphate (P_i_), and O_2_). If the oxygen demand cannot be met, pyruvate is converted into lactate. Under these anaerobic circumstances, no more ATP is produced, but the reducing equivalent nicotinamide adenine dinucleotide (NAD) is regenerated to become available for a new passage of glycolysis.

Glucose homeostasis is dependent on the ingestion of carbohydrates, metabolization of glycogen (=glycogenolysis), and hepatic de-novo-synthesis (=gluconeogenesis) of glucose. These metabolic pathways are mainly regulated by two hormones. Insulin lowers elevated blood glucose levels and allows the intracellular metabolization of glucose, while glucagon acts as the counterpart and increases blood glucose level.

While performing exercise, blood glucose level decreases [85] because of increased glucose uptake in skeletal muscle [86]. Indeed, the insulin-regulated induction of glucose transporter type 4 (GLUT4) in the muscular surface membrane enables a 20–30-fold increased uptake in muscle [138]. To maintain normoglycemic levels, the liver glucose output increases three-to fivefold during exercise [139]. Initially, stored glycogen is metabolized via glycogenolysis. Once the glycogen stores are depleted, increased gluconeogenesis may counteract the imminent drop in blood glucose levels [86,94]. By increasing glucose utilization and uptake, exercise successfully improves hyperglycemia. Further, there are numerous reports showing that regular exercise in patients with diabetes improves insulin sensitivity [140,141], reduces glycosylated hemoglobin (A1C) [142], and decreases fasting insulin levels as well as fasting blood sugar [143], especially when physical activity is combined with healthy diet [144].

Interestingly, the time of activity seems to play an important role, as glucose tolerance, insulin sensitivity, and the oxidative capacity of skeletal muscle undergo circadian oscillations displaying a day–night rhythm [145]. In lean, healthy subjects, peak oxidative capacity and highest resting energy expenditure coincide at the end of the day [95]. Concordantly, it was recently shown that in individuals with type 2 diabetes, afternoon exercise was more effective in lowering blood glucose levels than morning exercise, which in fact had deleterious effects [96]. Clearly, more research is needed on this topic, but it would be beneficial to identify an optimal time frame for exercise.

#### 3.1.3. Immunometabolism

Immunometabolism is a recently explored field that unravels intracellular metabolic pathways in immune cells, which can lead to functional changes [146]. Indeed, inflammatory cells, just like the skeletal muscle mentioned initially, require ATP as a fundamental fuel for growth, differentiation, and function.

Macrophages, for example, virtually devour glucose to generate the required energy for cytokine production and phagocytosis [147,148]. The traditional view that oxygen supply is the sole determinant of the prevailing glucose metabolism, i.e., oxidative phosphorylation or glycolysis, is now considered outdated. Rather, the metabolic state is dictated by an interplay of numerous metabolic and immunological factors that are interwoven and do not follow an “all-or-nothing” principle. Changes in metabolic pathways are closely linked to effector functions of the immune system, particularly in the production of various cytokines [146]. Macrophages metabolize glucose differently depending on their phenotype. Glycolysis can be activated quickly. Clearly, in the tough fight against invading pathogens, there should be no delay. Therefore, inflammatory macrophages (M1) are more likely to switch to glycolysis. In contrast, oxidative phosphorylation is a more complex and therefore slower process, as several mitochondrial enzymes must first be activated. Oxidative phosphorylation is mainly used by reparative macrophages (M2) [149,150,151].

The transcription factor hypoxia-inducible factor (HIF)-1, whose discovery was recognized with the Nobel Prize in Physiology or Medicine in 2019 [152], represents a compelling interface between the immune system and metabolism. HIF-1 consists of two subunits, the oxygen-sensitive HIF-1α and the constitutively expressed HIF-1β [153]. HIF-1 primarily serves the adaptation of the organism during deprived oxygen supply, for example by inducing glycolysis-associated enzymes for ongoing ATP synthesis [154,155]. The crux is that the expression of HIF is not only regulated by oxygen availability but also by other (atherosclerosis-associated) stimuli. For example, activation of TLR, particularly TLR4, leads to a switch from oxidative phosphorylation to glycolysis in immune cells via increased expression of HIF [156]. This step seems contradictory from an energy balance point of view—after all, only two molecules of ATP can be generated in this way—but turns out to be a smart move, since the induction of glycolysis occurs more rapidly than oxidative phosphorylation thereby enabling the immediate supply of energy [146]. Thus, inflammation-triggering stimuli enhance HIF-regulated glycolysis, which propels the secretion of pro-inflammatory cytokines in macrophages through increased ATP production. This mutually reinforcing loop vividly demonstrates how closely metabolism and the immune system are connected and should, therefore, not be viewed separately, but as an interacting and integrative system.

HIF-1 also plays an important role during physical activity, where its expression has been mainly investigated in the skeletal muscle. Here, an acute bout of vigorous exercise promotes the up-regulation of HIF-1α due to increased oxygen consumption [153]. Regular training, however, encourages remodeling processes to guarantee a better aerobic performance, which includes increased capillaries and enhanced mitochondrial enzyme activity. Consequently, regular training decreases HIF-1α in skeletal muscle, correlating with better performance [106,157]. In atherosclerosis, HIF does not only fulfill the role of a culprit. One must clearly differentiate between systemic and local effects. While the systemic effects of HIF in the atherosclerosis context are beneficial, for example by stimulating the formation of collateral vascular networks, HIF locally has a disease-promoting effect through its influence on EC, vascular smooth muscle cells, and macrophages. Interestingly, studies in which the local influence of HIF was specifically inhibited, showed a reduced plaque size [158].

#### 3.1.4. Inflammation

##### Production

Circulating blood leukocytes serve as a risk factor and prognostic indicator for atherosclerosis [159,160]. As monocytes and neutrophils have a short life span (<72 h), their abundance in circulation, as well as their recruitment to cardiovascular organs, highly depends on their production rate [161], which may be used therapeutically [162].

While a number of cardiovascular risk factors increase hematopoiesis [163], recently published data showed that regular exercise decreases leukocyte production. Hematopoietic stem and progenitor cells (HSPCs) give rise to mature leukocytes in response to cytokines, growth factors, and danger signals. The maintenance and regulation of these stem cells are largely instructed by the bone marrow niche, a local tissue microenvironment consisting of stromal bone marrow cells [164,165]. As displayed in Figure 2, regular physical exercise decreases HSPC proliferation by inducing quiescence-promoting niche factors. Specifically, exercise decreases the production of adipose tissue-derived leptin, a pro-inflammatory adipokine [166], thereby inducing the expression of quiescence and retention factors in leptin receptor-positive stromal bone marrow cells. Consequently, decreased hematopoiesis prevents the expansion of leukocytes in aortic plaques and reduces plaque size in mice [99]. In concert with these findings, there are several human studies demonstrating that moderate exercise diminished circulating leukocyte counts [97,98].

Inflammatory diseases such as atherosclerosis additionally promote extramedullary hematopoiesis. HSPCs relocate from the bone marrow to the spleen [167,168], where they differentiate into Ly6C^high^ monocytes. These monocytes enter the circulation, locally accumulate in atherosclerotic plaques, and contribute to disease progression by differentiating into macrophages, which eventually will form foam cells [169]. So far, it is not clear whether exercise also affects extramedullary hematopoiesis. Splenic monocytes egress in response to adrenergic signaling [170]. It would be interesting to see whether exercise somehow affects splenic release, taking into account that chronic exercise correlates with increased vagal tone [161,171].

##### Recruitment

During an acute bout of exercise, circulating leukocyte numbers are elevated. It is well-studied that exercise acutely promotes the release of leukocytes, which consequently enhanced re-deployment and re-distribution [12,41]. However, it is not known whether rapidly released cells also increasingly enter atherosclerotic plaques, which indeed represents a compelling destination [72]. Knowing that exercise reduces inflammatory cells in plaques [99], it would be interesting to see whether altered recruitment capacities are also accountable.

##### Plaque Behavior

Recruited cells fulfill different functions.

Monocytes are central regulators in atherosclerosis. Once migrated, they develop into macrophages. By provoking lipid retention, macrophages elicit a strong myeloid cell response with pro-inflammatory cytokine secretion, in situ macrophage proliferation, and further recruitment of myeloid cells [172,173]. Even though it has been shown that exercise reduces the proliferation of macrophages in adipose tissue [99], there is no evidence demonstrating that exercise affects macrophage proliferation in plaques. However, it is well-described that exercise reduces LDL [174], and decreased LDL levels account for reduced in situ macrophage proliferation [175] (Figure 3). To promote inflammation and exert pro-atherogenic functions, macrophages are, as mentioned above, equipped with TLR. Murine and human atherosclerotic lesions show enhanced TLR expression [176], which can be diminished by exercise. Specifically, exercise reduces TLR2 and 4. It is well-demonstrated that the reduction in TLR may counteract macrophages’ pro-atherogenic role: the depletion of TLR2 leads to reduced atherogenesis and a more stable plaque phenotype [177]. Additionally, TLR4 deficiency leads to decreased development of atherosclerotic lesions with reduced infiltration of monocytes [178] and smaller necrotic core areas [179].

Further, atherosclerotic lesions include helper and cytotoxic T cells, which secrete cytokines in response to antigens of cholesterol-rich lipoproteins [180]. T_H_1 cells are considered to be pro-atherogenic, as they activate macrophages by IFN-γ production. Contrary, T_REG_ cells have an atheroprotective role by promoting plaque stability via the production of IL-10 and transforming growth factor (TGF)-ß [159,180]. T_REG_ cells not only produce anti-inflammatory cytokines [181] but also inhibit pro-atherogenic T_H_1 and CD8+ cells [182], a fact that can be leveraged therapeutically. Indeed, the elevation of T_REG_ numbers via adoptive transfer reduced plaque size in atherosclerotic mice [183], while depletion of T_REG_ cells exacerbated atherosclerosis [184]. Several studies provided evidence that chronic moderate exercise elevates the number of immunomodulatory T_REG_ cells [38,39]. However, the exact mechanism has not yet been identified. One possible explanation would be that exercise decreases leptin, which promotes T_H_1 and T_H_17 differentiation [185,186] and decreases T_REG_ proliferation [187,188]. Another explanation would be that the increase in T_REG_ cells is mediated via vitamin D. As mentioned earlier, exercise elevates VDR on T lymphocytes [15] and vitamin D, subsequently, increases the number of T_REG_ cells [189].

For intercellular communication, the secretion of interleukins is essential. IL-6 has a prominent, albeit ambivalent, role in this process. On the one hand, IL-6, typically following IL-1 stimulation, ignites inflammation in atherosclerosis. Its pro-atherogenic role in CVD makes IL-6 a suitable pharmacological target, which is indeed already being evaluated in clinical trials [190,191]. On the other hand, skeletal muscle-derived IL-6, which is released during exercise, promotes anti-inflammatory properties by inducing IL-10 and interleukin-1 receptor antagonist protein (IL-1ra). IL-10 suppresses pro-inflammatory T_H_1 cells, while IL-1ra intercepts the pro-atherogenic effects of IL-1ß [29,192]. The relationship between IL-6, exercise, and atherosclerosis remains largely untouched and urgently requires further research.

### 3.2. The Consequences of Exercise-Related Risk Modulation on the Progression of Atherosclerosis

The clinical outcome and prognosis of atherosclerosis are mainly influenced by plaque size, i.e., the extent of lumen narrowing, and plaque composition determining the risk of plaque erosion/rupture. While plaque size, depending on its location, is an easy parameter to assess, plaque stability is somewhat more complicated to evaluate and relies on several factors. The atherosclerotic plaque is encased by a fibrous cap, which wraps around the plaque like a protective shield [69]. While smooth muscle cells establish the fibrous cap by producing an extracellular matrix (ECM) [193], matrix metalloproteinases (MMPs) promote their degradation through proteolytic action. The activity of MMPs is physiologically inhibited by tissue inhibitors of matrix metalloproteinases (TIMPs) [194].

The unstable “vulnerable” plaque is mainly characterized by a thin fibrous cap caused by the scarcity of smooth muscle cells and a disequilibrium between MMPs and their endogenous TIMPs. Further, the progressive increase in foam cells increases susceptibility to plaque rupture [195]. Several studies reported that exercise reduces plaque size in both early- and late-stage atherosclerosis [100,101,102], illustrating that exercise suppresses the disease nourishing components, such as impaired metabolism and inflammation. Further, atherosclerotic lesions in exercise-trained mice contain more collagen and elastin [100,101], which form the two major components of the ECM and are crucially important for the maintenance of the structural integrity of the vascular wall [196]. Therefore, exercise does not only reduce plaque size but also promotes plaque stability, which ultimately improves outcomes. To date, the extent to which MMPs or TIMPs are affected by exercise is not clear because studies are conflicting [100,101,102], necessitating further research to fill this gap.

## 4. Conclusions

Impaired metabolism and persistent inflammation strongly nurture the unrelenting process of propelling the progression of life-threatening atherosclerosis. In the battle against these causal contributors, exercise represents an ingenious weapon, as both metabolism and inflammation are powerfully combated. Exercise increases energy consumption and ameliorates substrate metabolism in numerous ways. Additionally, exercise creates an anti-inflammatory environment by diminishing the number of circulating disease-propagating leukocytes and eliciting several phenotypic alterations. Taken together, exercise represents a simple but potent preventive and therapeutic strategy to beneficially modify the course of atherosclerosis.

## 5. Future Directions

In a world that is spinning faster and increasingly trimmed to efficiency, a challenging, albeit very exciting goal is the improvement of training timing, duration, and intensity. However, one has to take into account that exercise cannot be performed all day long, but rather during short fractions of a day. The use of new devices, i.e., smart watches, allows for the accurate recording of health data and daily summation of physical activity by capturing and recording movement throughout the day, whether one is moving, exercising, or standing. The possibility to now easily detect health data will allow us to measure the effects of “easily to integrate into everyday life” physical activity. Just recently, it has been shown that three to four minutes of vigorous lifestyle physical activity (VILPA), such as very fast walking while commuting to work, is associated with lower all-cause CVD and cancer mortality risk [197]. Studies like this will help us to better comprehend “how little is enough” while still maximizing the beneficial effects of exercise. In addition, the metabolic and inflammatory facets of exercise need to be further characterized. The everlasting expansion of research methods in the field of genomics, proteomics, and lipidomics will enable us to accurately define the multiple effects of exercise, identify biomarkers, and develop new pharmaceutical targets on our way toward precision medicine.

**Literature Search:** We performed an exhaustive search in PubMed and Google Scholar using the keywords “exercise”, “inflammation”, “metabolism”, “physical activity”, “exercise immunology”, and “atherosclerosis”. We included studies published within the last 10 years from the most relevant journals and critically evaluated each article in terms of key results, limitations, and the suitability of the methods, as well as quality and interpretation of the results.

## Figures and Tables

**Figure 1 ijms-24-03394-f001:**
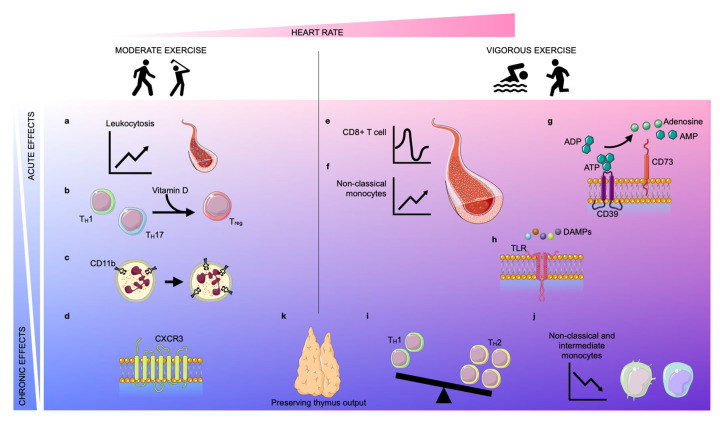
**Acute and chronic effects of moderate and vigorous exercise**. Depending on the increase in heart rate (pink area), exercise can be classified as moderate exercise or vigorous exercise. The effects of exercise can be classified as acute (light blue area) or chronic (blue area). Acute **moderate exercise** induces a leukocytosis in the blood (**a**) and a transient upregulation of the intracellular vitamin D receptor (VDR) in T lymphocytes, inducing T cell phenotype switching (**b**). Further, moderate exercise enhances the expression of CD11b on neutrophils acutely and chronically (**c**). In the long term, physical exercise improves the trafficking behavior of both CD8+ cells and natural killer (NK) cells by regulating the expression of CXCR3 (**d**). **Vigorous exercise** induces alterations in lymphocyte numbers in a biphasic way: an instantaneous lymphocytosis is followed by transient lymphopenia (**e**). Further, non-classical monocytes are increased during exercise (**f**). Vigorous exercise acutely elevates the expression of CD39 and CD73 on CD4+ T lymphocytes which assist in the conversion of adenosine triphosphate (ATP)/adenosine diphosphate (ADP) into adenosine monophosphate (AMP)/adenosine, respectively (**g**). Vigorous exercise reduces the expression of Toll-like receptors (TLRs) 1, 2, and 4 on CD14+ monocytes in the early post-exercise period, inducing anti-inflammatory effects (**h**). In the long term, exercise also promotes anti-inflammatory effects by encouraging anti-inflammatory phenotype switching in lymphocytes and monocytes. The balance between T_H_1 and T_H_2 cells shifts in favor of T_H_2 (**i**) and the frequency of intermediate and non-classical monocytes decreases due to diminished expression of CD16 (**j**). Finally, both moderate and vigorous exercise preserve thymic output (**k**).

**Figure 2 ijms-24-03394-f002:**
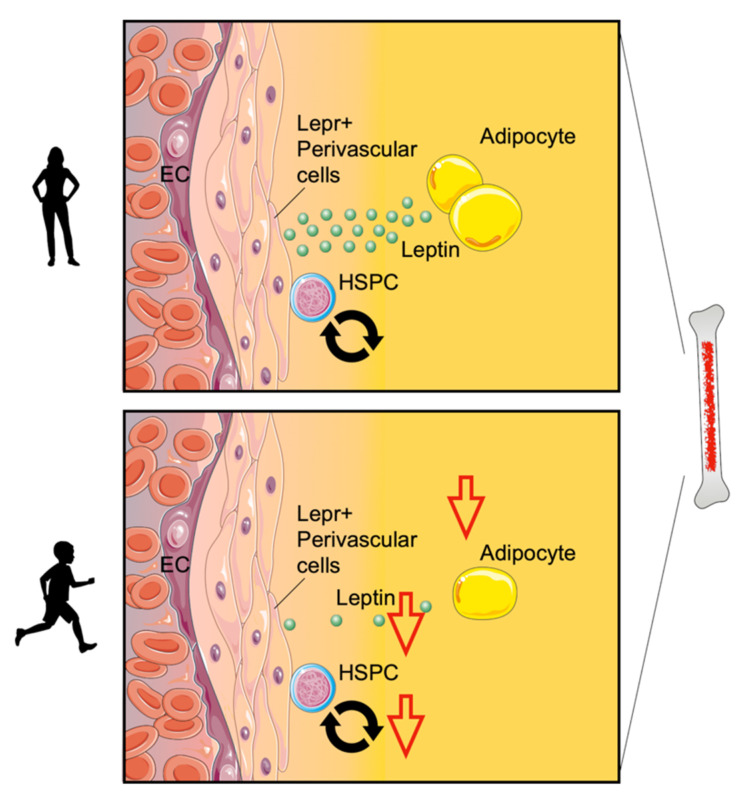
**Exercise modulates bone marrow hematopoiesis.** Physical exercise decreases hematopoietic stem and progenitor cell (HSPC) proliferation by inducing quiescence-promoting niche factors. Specifically, exercise decreases the production of adipose tissue-derived leptin and induces the expression of quiescence and retention factors in leptin receptor-positive stromal bone marrow cells (Lepr+ perivascular cells).

**Figure 3 ijms-24-03394-f003:**
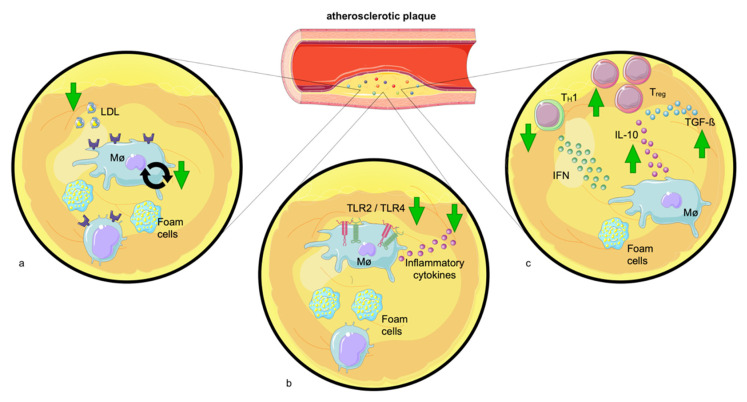
**Exercise attenuates the pro-inflammatory environment of atherosclerotic plaques.** (**a**) Exercise lowers LDL levels and may hence decrease LDL-induced plaque macrophage proliferation. (**b**) Physical exercise decreases Toll-like receptors (TLRs) −2 and −4 levels. The reduction in TLR expression diminishes the release of pro-inflammatory cytokines by plaque macrophages. (**c**) Exercise elevates the number of immunomodulatory T_REG_ cells which promote plaque stability producing interleukin-10 (IL-10) and transforming growth factor (TGF-β). Additionally, physical exercise reduces the quantity of T_H_1 cells which activate macrophages by interferon-gamma (IFN-γ) production.

**Table 1 ijms-24-03394-t001:** Overview of human trials cited in the text.

Study Population	CVRF	Intervention	Comparison	Main Results	Reference
**10 male** (28 ± 2 yrs)	-	4 h of treadmill exercise 1 h of recovery	active vs. inactive	**active:** increased (↑) triglyceride oxidation during exercise decreased (↓) glycerol and free fatty acid rate of appearance in plasma during recovery	[83]
**5 individuals** (26 ± 1.9 yrs)	-	4 h of treadmill exercise at 40% VO_2_ max 2 h of recovery	before and after	**after:** ↑ lipolysis	[84]
**7 male** (25 ± 3 yrs)	-	4 h of bicycle exercise at 57% VO_2_ max	before and after	**after:** ↑ plasma glycerol and free fatty acids ↓ intramuscular triacylglycerol and glycogen	[85]
**6 male** (28 ± 3 yrs)	-	4 h of bicycle exercise at 30% VO_2_ max	before and after	**after:** ↓ blood glucose ↓ hepatic glucose output ↑ hepatic glycogenolysis ↑ splanchnic gluconeogenesis	[86]
**13 male** (25.3 ± 1.2 yrs)	-	6 weeks of intermittent training with a one-legged knee extension ergometer at 150% VO2 max	trained vs. untrained leg (intraindividual analysis)	**trained leg:** ↑ capillary growth ↑ proliferating endothelial cells	[87]
**6 male** (23 ± 2 yrs)	-	8 weeks of intermittent training with a one-legged knee extensor at 65% VO_2_ max	trained vs. untrained leg (intraindividual analysis)	**trained leg:** ↑ LPL activity ↑ capillary density ↑ VLDL uptake ↑ production of HDL cholesterol	[88]
**16 individuals** (25 ± 5 yrs)	-	90 min of bicycle exercise at 50% VO_2_ max	before and after	**after:** no change in LPL activity no association between LPL activity and VLDL oxidation rate	[89]
**10 male** (unknown age)	-	20 km run after overnight fasting	before and after	**after:** ↑ LPL activity ↑ plasma glucagon ↓ plasma insulin	[90]
**8 male** (26.9 ± 4.1 yrs)	-	90 min of bicycle exercise at 70% VO_2_ max followed by 1 min of full-effort sprints	before and after	**after:** ↓ VLDL no effect on LPL activity or lipoprotein concentration	[91]
**16 individuals** (45 ± 2.5 yrs)	**↑**	**training session:** 12 weeks of endurance exercise training, 3 times/week; **single exercise**: a single exercise session on a treadmill at 70% VO_2_ max	before and after	**training session:** ↑ PPARα, AMPKα, CPT I, and COX-IV protein content in skeletal muscle **single exercise:** ↑ PPARδ, PGC-1α, FAT/CD36, LPL protein content in skeletal muscle	[92]
**7 male** (64 ± 5 yrs)	-	6 weeks of bicycle exercise training at the lactate threshold level for 60 min/day, 5 times/week	before and after	**after:** ↓ body fat, fasting levels of plasma glucose ↑ HDL, LCAT ↑ fast-to-slow fiber-type transition	[93]
**6 male** (24 ± 1 yrs)	-	2 h of bicycle exercise at 60% VO_2_ max before and after completion of a strenuous endurance training program (12 weeks of cycle ergometer training at 75–100% VO_2_ max for 45–90 min/day, 6 days/week)	before and after	**after:** ↓ rate of glucose appearance ↓ hepatic glycogenolysis ↓ hormonal response to exercise ↓ availability of gluconeogenic precursors	[94]
**12 male** (22.2 ± 2.3 yrs)	-	within 24 h, following a standardized living protocol with regular meals, physical activity, and sleep		peak energy expenditure at 11 pm lowest energy expenditure at 4 am rhythmicity in mRNA expression of molecular clock genes in human skeletal muscle	[95]
**11 male** (60 ± 2 yrs)	**↑** diabetes type 2	2 weeks of afternoon high-intensity interval training (HIIT), 2 weeks wash-out period, 2 weeks of morning HIIT	morning training vs. afternoon training	**morning training:** ↑ glucose concentration ↑ TSH **afternoon training:** ↓ glucose concentration ↑ TSH ↓ T_4_	[96]
**42 female** (53 ± 9.8 yrs)	**↑**	6 weeks of bicycle exercise training, 30–60 min/day, 1–6 times per week	before and after	**after:** **↓** blood leukocytes	[97]
**16 individuals** (64 ± 6 yrs)	**↑**	wearing a device with vibration feedback to promote physical activity for a period of 16 weeks	before and after	**after:** ↓ production of interleukin-1ß, -8, and -10 ↓ glycolysis in peripheral blood mononuclear cells no effect on monocyte counts	[98]
**99 individuals** active (70 ± 6 yrs) vs. inactive (75 ± 5 yrs)	-	activity based on scoring parameters	active vs. inactive	**active:** ↑ CD4/CD8 ratio ↑ naïve T lymphocytes	[38]
**30 individuals** active (25 ± 1.4 yrs) vs. inactive (26.1 ± 1.9 yrs)	-	a single bout of high-intensity interval training on a motorized treadmill	active vs. inactive	**active:** ↑ CD4+ T cell frequencies immediately after training ↑ regulatory T cells (T_REG_)	[39]
**35 mal** (44 ± 17 yrs)	-	60 min bicycle exercise at 55% VO_2_ max	before and after	**after:** ↑ VDR expression in T cells	[15]

Note: increased (↑), decreased (↓), CVRF (cardiovascular risk factors), LPL (lipoprotein lipase), VLDL (very low-density lipoprotein), HDL (high-density lipoprotein), VO_2_ max (maximal oxygen consumption), PPAR (peroxisome proliferator-activated receptor), PGC-1α (peroxisome proliferator-activated receptor gamma coactivator 1-alpha), FAT/CD36 (fatty acid translocase), AMPKα (AMP-activated protein kinase), CPT I (carnitine palmitoyltransferase I), COX-IV (cytochrome c oxidase 4), LCAT (lecithin–cholesterol acyltransferase), TSH (thyroid-stimulating hormone), T_4_ (thyroxine), VDR (vitamin D receptor).

**Table 2 ijms-24-03394-t002:** Overview of animal trials cited in the text.

Animal	Intervention	Comparison	Main Results	Reference
** *ApoE^−^* ^/−^ ** **mice**	6 weeks of voluntary running	active vs. sedentary	**active:** decreased (↓) hematopoiesis in bone marrow ↓ leukocytes in atherosclerotic plaque ↓ proliferation of macrophages in adipose tissue	[99]
** *ApoE^−/−^* ** **mice**	10 weeks of voluntary running commencing at 12 weeks of age (early atherosclerosis) or 40 weeks of age (late atherosclerosis)	active vs. sedentary	**active**: ↓ lipid levels ↓ plaque stenosis, increased (↑) collagen and elastin ↑ MMP-2 ↓ TIMP-1 no differences in TIMP-2	[100]
** *ApoE^−/−^* ** **mice**	8 weeks of exercise training (EX) on a treadmill after 16 weeks of high fat diet (HFD); 8 weeks of exercise training on a treadmill after 16 weeks of HFD + atorvastatin (AT, 10 mg/kg/d) treatment (AT + EX); 8 weeks of atorvastatin (10 mg/kg/d) treatment after 16 weeks of HFD (AT)	active (±atorvastatin treatment) vs. sedentary (±atorvastatin treatment)	**all intervention groups:** ↓ plaque stenosis ↑ collagen and elastin ↓ MMPs ↓ TIMP-2 **EX:** ↑ TIMP-1 within lesions	[101]
** *ApoE^−/−^* ** **mice**	6 weeks of daily training on a treadmill after 16 weeks of HFD	active vs. sedentary	**active:** ↓ plaque stenosis ↑ collagen and elastin ↓ MMP-9 ↑ TIMP-1 no changes in body weight or lipid levels	[102]
**Wistar rats**	18 weeks of daily swimming training for up to 6 h	active vs. sedentary	**active:** ↑ HSL enzyme activity in adipose tissue ↑ HSL protein concentration in adipose tissue no changes in HSL enzyme activity in skeletal muscle or HSL protein concentration in adipose tissue	[103]
**Wistar rats**	9 weeks of treadmill running in the early/late part of the active phase	early vs. late activity	**late activity:** ↑ lipolysis ↑ HSL expression ↑ sensitivity to ß-adrenergic signaling	[104]
**C57bl/6 mice**	HFD HFD + regular exercise for 10 weeks (HFD + EX)	standard diet (S)	**HFD compared to S:** ↓ carnitine concentrations ↓ mRNA and protein levels of genes involved in carnitine synthesis **HFD + EX compared to S:** no difference between HFD + EX and S group	[105]
**C57bl/6 mice and *HIF-1α*^−/−^ mice**	6 weeks of treadmill running for 30 min/d, 5 days per week; endurance test with treadmill running until exhaustion	C57bl/6 mice vs. *HIF-1α^−/−^* mice	**C57bl/6 mice after 6 weeks of training:** ↑ oxidative capacity **untrained *HIF-1alpha^−/−^* mice:** ↑ capillary to fiber ratio ↑ oxidative enzyme activities ↓ pyruvate dehydrogenase kinase expression in muscle (=adaptive response in skeletal muscle to endurance training)	[106]

Note: increased (↑), decreased (↓) MMPs (matrix metalloproteinases), TIMP (tissue inhibitor of metalloproteinases), HFD (high-fat diet), HSL (hormone-sensitive lipase)

## Data Availability

Not applicable.

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
