# Peer review of "The Impact of Exercise on Immunity, Metabolism, and Atherosclerosis"

_ijms, 2023, doi:10.3390/ijms24043394_

Round 1
Reviewer 1 Report
This review brings a little novelty, as it discusses the aspects of exercise related to the disease, but emphasizing the mediation of inflammatory biomarkers. Despite the theme's absence, I've included some thoughts below.
1. I think it would be interesting to make it clear that this is a narrative type review. I know that this type of review is quite common in the areas of molecular biology, biochemistry, among others. However, I think that some aspects can be added and this is one of them.
2. Make clear, in the abstract and at the end of the article introduction (mostly) a rationale for the importance of this article and its related topic.
3. Pay attention to the use of the terms physical activity and physical exercise. It can confuse the reader (especially those who are not from the area) to understand the terms as having the same meaning.
4. The summary describes what was done in the review, however, I think that at the end of it, some indications for future studies can be given, some topic that involves the study of the relationships between exercise, inflammation and arteosclerosis that has drawn attention, etc.
5. The caption for figure 2 is too long. I think there is some text there that can be transferred to the body of the text. Pay attention to the resolution of the figures in the final version of the article. "We" are using programs or image banks that do not give the figures an adequate resolution. The publisher won't call you attention for this, and I think the all figures is too interesting to be published in low resolution.
5. Even though it is a narrative review, I would like it to be clear to the reader how the reasoning for seeking information was developed.
6. Are you guided by terms or one by one theme in particular? How did you arrive at the main terms (which can be seen as the skeleton that guided the review)? Is it a topic of interest? Is it part of a project and this text is a preliminary work?
7. I think that a PICOS table (https://libguides.mssm.edu/ebm/ebp_pico) with the main articles found and cited in the text, which go against the theme, can be added to this text. There are people who will only read the abstract, but will be interested in instantly seeing the studies found related to the topic.
8. Maybe I can give some emphasis to interleukins. Talking about IL-10, however, when talking about inflammation and exercise, it would be interesting to give some emphasis to the biomarkers of this family.
9. Finally, this paper only enriches the literature if it leaves a strong message, opening a precedent of investigation for future studies. Strengthen this point at the conclusion of the article please.
Reviewer 2 Report
The manuscript explores the relationship between exercise immunology, metabolism, and atherosclerosis. The relationship between metabolism and immune system cells is known and the object of several studies. However, the literature is scarce when the subject is immunometabolism and exercise. Atherosclerosis, a condition of metabolic imbalance, can play an essential role in the consolidation of a vicious cycle that includes cells of the immune system and inflammation. The topics of the manuscript were well-defined and presented in the text in a logical and didactic sequence. However, the relationship between topics, especially in the context of physical exercise, is the manuscript's weak point and needs improvement. For example, exercise can alter metabolic sensors in leukocytes, changes in serum and muscle metabolites, and inflammatory mediators. How does this modulation occur in atherosclerotic conditions and vice versa? The lack of relationship between the review themes is discouraging, but the interweaving of contents can make the manuscript more attractive for scientific literature and readers. The authors should also explore new hypotheses. What future research will the manuscript support on immunometabolism, exercise and atherosclerosis? Finally, the authors must inform the strategy for selecting the papers read to build the review.
Round 2
Reviewer 1 Report
Thank you very much for your effort to raise the quality of the article. I recommend just putting the "conclusion" section last.
Reviewer 2 Report
The authors made profound changes to the manuscript. Questions were answered and suggestions were satisfactorily considered. The current version of the text will contribute to the literature. Therefore, I believe that the manuscript is ready to be published.